# Current and Future Tools for Diagnosis of Kaposi’s Sarcoma

**DOI:** 10.3390/cancers13235927

**Published:** 2021-11-25

**Authors:** Nicolas Dupin, Aude Jary, Samia Boussouar, Charlotte Syrykh, Amir Gandjbakhche, Sébastien Bergeret, Romain Palich

**Affiliations:** 1Dermatology Department, Cochin Hospital, AP-HP, Institut Cochin, INSERM 1016, Université de Paris, 75014 Paris, France; nicolas.dupin@aphp.fr; 2Virology Department, Pitié-Salpêtrière Hospital, AP-HP, Pierre Louis Epidemiology and Public Health Institute (iPLESP), INSERM 1136, Sorbonne University, 75013 Paris, France; aude.jary@aphp.fr; 3Cardiothoracic Imaging Unit, Pitié-Salpêtrière Hospital, AP-HP, ICAN Institute of Cardiometabolism and Nutrition, INSERM, Sorbonne University, 75013 Paris, France; samia.boussouar@aphp.fr; 4Department of Pathology, University Cancer Institute of Toulouse-Oncopole, 31000 Toulouse, France; charlotte.syrykh@iuct-oncopole.fr; 5Section on Analytical and Functional Biophotonics, National Institute of Child Health and Human Development, National Institutes of Health, Bethesda, MD 20892, USA; gandjbaa@mail.nih.gov; 6Nuclear Medicine Department, Pitié-Salpêtrière Hospital, AP-HP, Sorbonne University, 75013 Paris, France; sebastien.bergeret@aphp.fr; 7Infectious Diseases Department, Pitié-Salpêtrière Hospital, AP-HP, Pierre Louis Epidemiology and Public Health Institute (iPLESP), INSERM 1136, Sorbonne University, 75013 Paris, France

**Keywords:** Kaposi’s sarcoma, diagnostic, staging, FDG-PET, non-invasive imaging tools

## Abstract

**Simple Summary:**

Kaposi’s sarcoma, a rare opportunistic tumor, is observed in four epidemiological conditions (AIDS-related, iatrogenic, endemic or classic KS). Although in most cases KS is an indolent disease, it can be locally aggressive and/or it can invade other organs than the skin, resulting in more severe presentations, especially in patients with severe immunosuppression. There is no consensus on the imaging workup that is necessary for either the initial staging of the disease or the follow-up. Future perspectives include the use of certain non-invasive imaging tools that may help to evaluate the clinical response to treatment, as well as certain new histological markers that may help in guiding the treatment planning for this atypical neoplasm.

**Abstract:**

Kaposi’s sarcoma (KS) is a rare, atypical malignancy associated with immunosuppression and can be qualified as an opportunistic tumor, which responds to immune modulation or restoration. Four different epidemiological forms have been individualized (AIDS-related, iatrogenic, endemic or classic KS). Although clinical examination is sufficient to diagnose cutaneous lesions of KS, additional explorations are necessary in order to detect lesions involving other organs. New histological markers have been developed in recent years concerning the detection of HHV-8 latent or lytic proteins in the lesions, helping to confirm the diagnosis when it is clinically doubtful. More recently, the evaluation of the local immune response has also been shown to provide some guidance in choosing the appropriate therapeutic option when necessary. We also review the indication and the results of conventional radiological imaging and of non-invasive imaging tools such as ^18^F-fluoro-deoxy-glucose positron emission tomography, thermography and laser Doppler imaging for the diagnosis of KS and for the follow-up of therapeutic response in patients requiring systemic treatment.

## 1. Introduction

Kaposi’ sarcoma (KS) is an atypical tumor that is associated with immunosuppression. KS can be qualified as an opportunistic tumor, and it is well demonstrated that in patients with immunodeficiency, the restoration of immunity can be sufficient to cure KS. For example, in patients with acquired immunodeficiency syndrome (AIDS)-KS, antiretroviral treatment alone has been shown to cure KS [1], and since the development of combined antiretroviral therapies (cART), the incidence of KS has dramatically decreased in most of the countries where those antiretroviral agents are currently available [2,3,4,5].

There are four major epidemiological forms of KS: classic (or Mediterranean) KS (CKS), which mostly affects elderly men of Mediterranean, east European, or Jewish heritage with a peak incidence after the sixth decade of life; endemic African KS, which occurs in sub-Saharan Africa among children and adults; iatrogenic immunosuppressive KS, which affects patients who are receiving chronic immunosuppressive therapy, mostly organ-transplant recipients; and Epidemic KS or AIDS-KS, which affects people living with HIV (PLHIV), which is the most common AIDS-associated malignancy [6]. Amongst AIDS-KS, a sub-entity of KS has also emerged in the form of long-term HIV-aviremic PLHIV.

Cutaneous and mucosal manifestations, which are largely predominant in all types of KS, allow a generally obvious clinical diagnosis, since the lesions are characteristic, although they may vary in aspect. The diagnosis is based on the combination of different clinical elements: purple to brown macules (Figure 1A), a distribution of lesions along tension lines, a yellow-green color or contusiform appearance on the peri-lesion, lymphedema, and disseminated lesions. The initial presentation is typically that of purple macules or papules progressing secondarily to plaques or nodular lesions (Figure 1B). The preferred skin sites are the lower limbs, the feet and the trunk with occasional extension to the upper limbs, the hands, the head and the neck. Oral (Figure 1C) and genital lesions can be observed in some instances, especially in patients with AIDS-KS. However, cutaneous and mucosal lesions are not always present or isolated. Lymphatic involvement is often associated with mucocutaneous involvement, manifesting as localized lymphedema which may result in complications such as pain, maceration and ulcerations (Figure 1D). In HIV-infected children, KS lymphadenopathy is more common than skin lesions. In some cases, especially in immunocompromised patients, extra-cutaneous localizations are observed, particularly in lymph nodes, the lungs and the digestive tract, the latter two of which may be life-threatening.

The detection of lesions in certain anatomical localizations, whether symptomatic or not, requires further explorations. A histological analysis, from bronchial or digestive endoscopy-guided biopsies, or from biopsies of lymph nodes, bone, or skin, allow for a definite diagnosis. Conventional imaging, including CT and MRI, can identify visceral and bone lesions that may go unnoticed upon clinical examination. ^18^F-fluoro-deoxy-glucose (FDG) positron emission tomography (PET), or FDG-PET, is also a promising examination, both for the diagnosis and follow-up of patients with KS. Tools that were developed for monitoring the treatment response of KS are also essential. Indeed, KS is a very polymorphic disease, for which the treatment response is very difficult to quantify. As such, it is complex to assess the effectiveness of therapies in clinical trials. New non-invasive imaging methods that are based on thermography and laser Doppler imaging (LDI) could revolutionize the initial staging of KS and the evaluation of the treatment response.

Exhaustive screening for KS lesions is crucial to establish the correct staging of the disease, which guides therapeutic choices, and allows for a prognosis assessment. Depending on the KS lesion localization, additional examinations are very variable, and it is generally necessary to combine several tools. Another challenging aspect is the assessment of the therapeutic response under cART in HIV+ patients and/or under specific treatment (cytotoxic drugs, antiangiogenic drugs, targeted therapies, immunotherapy, etc.). For skin lesions, this response is often difficult to determine, and there are now non-invasive imaging options that could help distinguish scarring from lesions that are still active.

That is why in this review we aim to describe the current and future tools for the diagnosis of KS.

## 2. Staging of Kaposi’s Sarcoma

As KS cannot be considered a true neoplasm, to date, the staging of KS has not been unified or incorporated into the tumor, node and metastasis (TNM) staging system developed by the American Joint Committee on Cancer (AJCC). Instead, the modified AIDS Clinical Trials Group (ACTG) staging classification, which is based on the tumor, immune status and systemic illness (TIS), is used for AIDS-related KS. In the era of cART, mostly the T and S stages (and not I) seem useful for the identification of patients with poor prognoses (Table 1) [7,8].

Outside the context of AIDS-KS, there is no consensual classification for the staging of classic KS, and there is no accepted classification for iatrogenic or endemic KS.

For a long time, the classification of classic KS relied on Krigel’s classification, which describes four stages: stage (I) is limited to the skin, localized, and without aggressive histological features; stage (II) is characterized by locally aggressive skin involvement with or without locoregional lymphadenopathy; stage (III) consists of generalized skin lesions and/or lymph nodes; and stage (IV) is visceral involvement [9]. In Krigel’s classification, two types (A and B) are described according to the presence or absence of general symptoms such as the loss of more than 10% of body weight and an unexplained fever lasting more than two weeks.

However, most recent reviews admit that the classification described by Brambilla et al. is probably more appropriate [10]. Based on a retrospective analysis of more than 300 cases of classic KS, they proposed the following classification, which differs from Krigels’ staging system: stage (I) is the macronodular stage, which is defined by small lesions (macules) confined to the lower extremities; stage (II) is the infiltrative stage, which consists of larger lesions (plaques) confined to the lower extremities; stage (III) is the florid stage, in which multiple larger lesions (plaques and nodules) are confined to the lower extremities; and stage (IV) is the disseminated stage, which is characterized by multiple large lesions extending beyond the lower extremities.

Although histopathological features are identical in different epidemiological forms of KS, prognosis varies largely between them. As such, it is also important to appreciate certain prognostic parameters which are not perfectly translated in the currently recommended staging classifications.

First, the overall tumor volume should be assessed, as well as the risk of visceral involvement. However, apart from AIDS-KS and some cases of iatrogenic KS, visceral involvement is very rare. KS is primarily a skin tumor, and the extension of the disease onto the skin in itself may constitute the best parameter for assessing disease severity. The degree of skin extension can vary from one lesion of less than 1 cm up to more than 100 plaques all over the skin surface. Evidently, the severity of KS is associated with the degree of skin extension. Yet, some forms of KS have a limited skin surface extension but are locally aggressive. This is not so rare in patients with endemic KS, in which there may be only one skin lesion that progresses to an ulcerative stage with exuberant proliferation and soft tissue invasion, or even invasion of the adjacent bone. In such cases, the amount of skin surface involved can be very limited and will not reflect the severity of the disease, and systemic treatment such as chemotherapy or immunotherapy is required.

Due to the absence of a consensual staging system for non-AIDS-KS, European consensus-based interdisciplinary guidelines were established, which state that patient management should distinguish three situations: localized non-aggressive, locally aggressive and disseminated KS [11]. These guidelines state that for CKS, endemic KS and post-transplant KS evaluation and staging, the workup should be discussed on an individual basis depending on the symptoms and the rate of lesion development, as presented in Table 2 [11].

For the screening of bronchial and digestive mucosal KS, an endoscopy is the best way to visualize and biopsy the macroscopic lesions in order to obtain a histopathological diagnosis.

Finally, if in most cases there are no histopathological traits that distinguish different forms of KS, there are yet some very rare cases of particularly aggressive tumors which are histopathologically qualified as anaplastic (Figure 1C), the prognosis of which is very poor [12].

## 3. Histopathological Characteristics

In order to definitely establish the diagnosis of KS, a histopathological analysis of a biopsy sample is required. Conventional hematoxylin and eosin staining show several features that are shared by the different epidemiological forms of KS (AIDS-related, iatrogenic, endemic or classic KS) with aspects depending on the nature of the clinical lesion as it progresses from the patch to the plaque and nodular phases [13,14]. Multiple clinicopathological forms of KS have been described but frequent overlap exists, and it is unclear whether these stages occur in sequential order. Among the usual variants, patch-stage KS is characterized by a proliferation of small and irregular endothelial-lined spaces surrounding normal dermal vessels and adnexal structures. The protrusion of native microscopic vascular structures into the lumens of dilated neoplastic channels may be seen and results in the characteristic but non-specific promontory sign (Figure 2A,B). This early stage may be the most difficult to histopathologically distinguish from the other conditions as it can mimic other inflammatory skin disorders such as minor vascular anomalies and inflammatory conditions. Plaque-stage lesions of KS show a more diffuse vascular proliferation throughout the dermis with fascicles of spindled cells and occasional extension into the underlying subcutis. Mitotic figures are sparse and there is no significant nuclear or cytological pleomorphism. Nodular KS exhibits dermal expansion by a relatively circumscribed, variable cellular proliferation of neoplastic spindled cells with mild to moderate cytologic atypia, separated by slit-like spaces containing erythrocytes (Figure 2C,D). All stages are accompanied by variable inflammatory lymphocytic and plasma cell infiltrates, extravasated erythrocytes, hyaline globules and hemosiderin-laden macrophages. Numerous other pathological variants of KS have been described depending on the presence of additional features or characteristic infiltration patterns: the anaplastic, lymphangioma-like, lymphangiectactic, bullous, telangiectatic, cavernous hemangioma-like, hyper-keratotic, keloidal, micronodular, pyogenic granuloma-like, ecchymotic, intravascular, glomeruloid and pigmented variants [13,14]. This clinicopathological classification carries no prognostic value except for anaplastic KS, which seems to be associated with aggressive forms [15]. Immunohistochemical staining using endothelial markers such as CD31, CD34, D2-40 and ERG are useful to differentiate KS from non-vascular spindle cell neoplasms (Figure 2E). The identification of HHV-8 by using LNA-1 antibody is the most effective immunostaining technique available to differentiate KS from its mimics (Figure 2F).

Immunomodulatory pathways could play an important role in KS pathogenesis. Over the past few years, several studies have therefore been carried out to investigate the prognostic value of programmed death-1 (PD-1)/PD-ligand-1 (PD-L1) expression in KS with controversial results and a lack of standardized protocols for PD-L1 detection [16,17,18,19]. The most recent studies, focusing on PD-L1 expression in the different cell populations, found a pathological-stage-related increase of PD-L1 expression in the tumor microenvironment [20,21,22]. These results provide the rationale for the clinical development of checkpoint inhibitors targeting PD-1/PD-L1 in refractory KS, but further clinical and molecular studies in large patient groups are needed, as well as standardization of PD-L1 testing.

## 4. Biological Diagnostic Tools

No current guidelines recommend the use of virological diagnosis tools to detect the HHV-8 genome or antibodies for the screening, monitoring or diagnosis of any epidemiological form of KS [11,23]. As such, there is no standardized method (target genes, PBMC, plasma or whole blood sample for peripheral compartment) or universal unit for techniques such as EBV quantification, which makes comparisons between studies difficult.

However, several studies have suggested that PCR-based methods, and mainly real-time polymerase chain reaction (PCR), may help to monitor KS. First of all, HHV-8 nucleic acid can be detected in several specimens including various biopsies, blood, broncho-alveolar lavage, ascites and pleural liquid with high sensitivity and specificity. Also, the HHV-8 genome has been detected in the specific HHV-8-infected “spindle cells” of all forms of KS [24]. Its detection by PCR can represent a valuable method for diagnosing KS, particularly in small skin-biopsy samples that might show histopathological overlap with other non-KS lesions [25]. The number of HHV-8-genome copies per cells in KS lesions may range from approximately 1 to 5 [26] and has been shown to increase with tumor burden in AIDS-patients compared to those with regressing KS [27]. In PBMC, patients with active KS also had a higher HHV-8-DNA viral load compared to patients with regressing KS [28].

Studies performed on whole blood or plasma samples reported variations in the HHV-8-DNA viral load in different subtypes of KS. In epidemic forms, the HHV-8-DNA viral load is undetectable in 7 to 21% of cases [29,30,31,32], whereas it seems to be less frequently positive in the classic and endemic forms (undetectable in 29 to 42% of cases) [30,33], depending on the studies. In AIDS-patients, the HHV-8-DNA viral load positivity and increase were correlated with the development of new KS lesions and disease progression [29,34,35,36]. In addition, studies have also shown a correlation between positive therapeutic responses and a reduction in the HHV-8-DNA viral load [37,38]. Patients with progressing KS had a 7.7-fold higher likelihood of having HHV-8 viral loads greater than 10,000 copies/10^6^ cells compared to patients with regressing KS [35]. In HIV-negative patients with KS, the HHV-8-DNA viral load was significantly lower compared to those with the epidemic KS form [28,30]. Furthermore, in classic and endemic KS, the disease progression and staging were significantly and independently associated with the positivity of HHV-8 viremia [33].

Various serology tests have been developed to detect lytic and/or latent HHV-8 antigens, and are based on immunofluorescence [39,40] (with or without previous stimulation of PEL cell lines), western-blot or enzyme-linked immunosorbent assay (ELISA) [40,41], but their usefulness in monitoring KS remains limited. So far, HHV-8 serologies are mainly available for sero-epidemiological studies or before solid organ transplantation, even though no recommendations exist concerning this point.

## 5. Radiological Presentation

Post-mortem studies suggest that more than a quarter of KS patients have visceral lesions commonly involving the lungs, liver, spleen and the digestive tract.

### 5.1. Pulmonary KS

KS may involve the lung parenchyma, airways, pleura, thoracic lymph nodes, or chest wall [42].

A chest X-ray may be normal or demonstrate nodular opacities and interlobular septal thickening with peribronchovascular predominance, pleural effusions and lymphadenopathy predominantly in the hilar regions [43]. The initial involvement develops in the perihilar regions and extends to the periphery. Lesions predominate at the base of the lungs (Figure 3).

CT findings include bilateral involvement in the mid and lower lung zones and, commonly, multiple bilateral flame-shaped nodular opacities [44,45]. The disease distribution is usually peribronchovascular with peribronchovascular interstitium and interlobular septal thickening caused by lymphatic obstruction or tumor invasion (Figure 4) [46]. Ill-defined nodular lesions along the lines of bronchovascular bundles can also be seen. Nodules can progress to consolidation. Ground-glass opacities (GGOs) can surround nodules and consolidations, thus forming the “halo sign” [47]. GGOs are caused by hemorrhage or exudation in an adjacent area [48]. A “crazy-paving” pattern can also develop. Dilated blood vessels in consolidations and nodules can be observed [49]. The obvious enhancement of nodules or consolidations may be explained by inflammation, vascular proliferation, and blood vessel dilation. Cavitation is reported in 8% of pulmonary involvement. The flame sign is a characteristic sign of pulmonary KS involvement. Cervical, axillary, mediastinal and hilar lymphadenopathy are reported in 30–50% of patients. Pleural effusion occurs in more than half of KS patients with lung parenchyma involvement [50]. Pleural effusion, often of low to moderate abundance, may be unilateral or bilateral, exudate or serohematic (Figure 5) [51], and rarely, some cases of nodular pleura have been described. Chylothorax can occur in around 20% of cases. Lymphatic involvement may be secondary to direct damage to the thoracic duct by tumor proliferation or secondary to extrinsic compression by mediastinal lymphadenopathy [52]. There are multiple cases in the literature of histologically proven KS-related pericardial effusions, commonly of moderate to large abundance, and of an exudative or hematic nature [53]. Chest CT manifestations change as the disease progresses. CT can follow the evolution of the disease under treatment, including the response to targeted therapy.

On magnetic resonance imaging, pulmonary involvement may present as irregularly increased signal intensity on T1-weighted images; reduced signal intensity on T2-weighted images (related to alveolar hemorrhage or fibrous components), as well as important signal enhancement after gadolinium injection, particularly in lesions with a peribronchovascular distribution. Those MR characteristics are highly suggestive of a KS diagnosis [45].

### 5.2. Hepato-Splenic KS

Hepato-splenic involvement is reported in approximately one-third of KS patients in autopsy case series (Figure 6).

Abdominal ultrasound imaging of the liver can demonstrate mild hepatomegaly with inhomogeneous cystic nodules along the peripheral branches of portal veins with hyper-echoic parenchymal bands. An abdominal CT may show inhomogeneous hepatomegaly with multiple small hypo-dense periportal nodules which may exhibit iso- or hyper-attenuation on delayed images, similar to that of hemangiomas [47]. MRI shows hyper-intense nodules on T1-weighted in-phase imaging and hypo-intense nodules on T1- weighted out-of-phase imaging. Splenomegaly and nodular hemangioma-like lesions are the most common forms of splenic involvement.

### 5.3. Other Visceral Involvement

Visceral involvement is often asymptomatic and can present as an enhanced mass with enhanced regional lymph nodes. Lymphadenopathies are common, especially in the retroperitoneum. The most frequently reported visceral complications are appendicitis, bowel obstruction (usually affecting the duodenum, secondary to submucosal tumors or diffuse wall thickening), hemorrhage, and perforation [42] (Figure 5 and Figure 6).

### 5.4. Musculo-Skeletal KS

Musculo-skeletal KS is rare. Lytic bone lesions and soft-tissue masses with cutaneous and subcutaneous involvement may be seen on conventional imaging [42,54]. The axial skeleton is more frequently affected and periosteal reactions are rare (Figure 7 and Figure 8).

## 6. FDG-PET

FDG-PET is a nuclear medicine imaging modality that uses FDG, which is a radio-labelled glucose analogue that is used for the exploration of cellular glucose metabolism. It is now systematically coupled with anatomical imaging (most often CT), for the anatomical localization of FDG uptake. FDG-PET is a well-recognized modality for the workup of various inflammatory and neoplastic diseases in which glucose metabolism is increased.

In KS, there is a known increase in cellular glucose metabolism which affects cells latently infected with HHV-8 [55], partly due to certain viral microRNA molecules [56]. Non-infected adjacent stromal cells may also be exposed to these microRNA via exosomal transfer from the infected cells, thus inducing an increase in glucose metabolism beyond the infected cancer cells [57]. As such, KS lesions may be highly avid for FDG, and FDG-PET can be used to detect KS lesions at various anatomical sites (Figure 9).

In the literature, several FDG-PET studies have reported KS patients with FDG-avid lesions, even though most of these are case reports. These studies have mostly concerned AIDS-related KS [58,59,60,61,62,63], but some have also concerned classic KS [64,65], transplant-related KS [66] and endemic KS [67].

FDG-avid KS lesions have been reported in the skin [58,59,62,64,67,68,69], lymph nodes [59,62,63,64,65,66,67], bone [61,64,67], lungs [59,61,62], gastro-intestinal tract [60], mouth [66], thyroid [66], and salivary glands [64], which demonstrates the ability of FDG-PET to detect KS lesions in various organs. Interestingly, for some of these authors, FDG-PET was instrumental in detecting unsuspected KS lesions [61,62,66]. One of the main advantages of FDG-PET compared to most conventional imaging studies is the whole-body exploration, which is of particular relevance in KS as it may affect a wide range of anatomical sites.

Only two patient series have reported FDG-PET results in KS. The first retrospectively evaluated the utility of FDG-PET for differentiating the diagnoses in 120 HIV patients with lymphadenopathy, only 13 of which had KS, and showed that FDG uptake in lymphadenopathy due to KS was of similar intensity to that of HIV-reactive lymphadenopathy, and lower than those due to lymphoma or mycobacterial and fungal infections [70]. The authors showed that FDG uptake with an SUV_max_ value greater than 10 excluded KS, but that the exact diagnosis of the lymphadenopathy could not be established based on PET alone (which is expected due to the non-specific nature of FDG uptake). However, this study outlined the value of FDG-PET for the localization of FDG-avid lymph nodes as targets for biopsy, which remains an important factor of obtaining diagnoses.

The second patient series is the only one to have reported the sensitivity and specificity of FDG-PET in a sizeable sample of KS patients (Pesque, et al., in preparation). This retrospective study, which included 75 patients with KS of all four subtypes, showed an excellent diagnostic performance of FDG-PET with sensitivity and specificity above 85% for the evaluation of KS involvement in the skin, lymph nodes, bone, muscle and lung. For the other localizations (digestive tract, liver, head and neck, mucosae), the specificity remained excellent (above 95%) but the sensitivity was lower, which was hypothesized to be due to small lesion size, as well as the physiological excretion of FDG in these sites, which may hinder lesion detection. Interestingly, the authors mention that in 24% of patients, FDG-PET revealed unknown lesions based on usual imaging workup, suggesting that FDG-PET may perform better than conventional imaging.

Some of the previously mentioned case reports included follow-up FDG-PET studies performed for treatment response assessment. In one case of gastric and mouth KS (without skin lesions), FDG uptake decreased almost completely in the gastric lesion following treatment with paclitaxel [60]. Another report of 3 KS cases with disseminated lymph node involvement, 2 of which had concomitant Castleman’s disease, showed a complete response on FDG-PET following highly active anti-retroviral therapy alone [63]. Another report of 2 cases of endemic KS with disseminated skin, lymph node and bone involvement showed a partial response on FDG-PET after treatment with anti-PD1 immunotherapy, concomitant to substantial clinical improvement [67]. Another case of concomitant KS and Castleman’s disease with disseminated lymph node involvement showed complete response on FDG-PET after treatment with steroid therapy and chemotherapy. These reports suggest that FDG-PET might also be an effective tool for the assessment of treatment response in KS, but further studies are warranted on the subject.

Overall, FDG-PET constitutes a promising non-invasive whole-body evaluation for detecting KS lesions. This modality seems effective for the identification of appropriate biopsy targets and may also be useful for staging, and possibly for the assessment of treatment response. However, available data mostly concerns case reports, and more studies are warranted to evaluate sensitivity and specificity and to determine the role of FDG-PET in relation to conventional imaging.

## 7. Non-Invasive Imaging Tools

Three non-invasive, non-contact imaging methods have either been investigated separately or in a multi-modality setting in order to assess the response to therapy of KS lesions.

The first study concerns the evaluation of two non-invasive methods, thermography and LDI, for their ability to quantitatively assess the parameters of vascularity in lesions of HIV-associated KS [71]. Thermography and LDI images of a representative KS lesion were recorded in 16 patients and compared to normal skin that was either adjacent to the lesion or on the contralateral side. Of the 16 patients, 11 had a temperature increase of greater than 0.5 °C and 12 had increased flux (blood flow measured by LDI) as compared to normal skin. There was a strong correlation between these two parameters (R = 0.81, *p* < 0.001). In ten patients, measurements were obtained prior to therapy and after receiving a regimen of liposomal doxorubicin and interleukin-12. After 18 weeks of therapy, the temperature and blood flow of the lesions were significantly reduced compared to the baseline (*p* = 0.004 and 0.002, respectively).

The third extensively evaluated technique is multi-spectral imaging, which collects images at several near-infrared wavelengths which are then inputted into a mathematical, optical skin model that considers the contributions from different analytes in the epidermis and dermis skin layers. Through a reconstruction algorithm, the percentage of blood in a given area of tissue and the fraction of that blood that is oxygenated are quantified, and can be compared to values that were previously measured in normal skin tissue either in the usual state or in a state of induced ischemia. [72]. This methodology has been applied in order to assess vascular KS lesions and the surrounding tissue before and during experimental therapies. Multi-spectral imaging has also been combined with LDI to gain additional information. The results indicate that these techniques are able to provide quantitative and functional information about tissue changes during experimental drug therapy and detect progression of diseases before changes are visually apparent [73,74]. However, one limitation inherent to this method is the time-consuming nature of the image reconstruction, which may limit time resolution of images. In order to achieve real-time imaging, one study reported the use of a reconstruction method based on principal component analysis for obtaining blood volume and oxygenation concentration maps [75]. These images were compared with clinical and pathological responses that were determined by conventional means. Another study using this method demonstrated that cutaneous KS lesions have increased blood volume concentration and that changes in this parameter are a reliable indicator of treatment efficacy, thereby differentiating responders and non-responders. Blood volume decreased by at least 20% in all of the lesions that responded by clinical criteria and increased in the two lesions that did not clinically respond. Responses that were assessed by multi-spectral imaging also generally correlated to the overall patient clinical response assessment, were often detectable earlier in the course of therapy, and were less subject to observer variability than conventional clinical assessment. Tissue oxygenation was more variable, with lesions often showing decreased oxygenation in the center surrounded by a zone of increased oxygenation [76].

## 8. Conclusions

KS is an opportunistic tumoral process which in most cases is an indolent disease and does not need specific treatment. However, in some instances, KS may be locally aggressive or disseminated, requiring a reliable assessment of the disease stage before choosing the appropriate therapeutic option. Among the non-invasive imaging methods, FDG-PET, thermography and LDI constitute promising tools for assessing the extension, disease activity as well as treatment response of KS lesions. In the future, the development of new histological parameters such as PD-L1 expression may also help to define the place of immune modulators in the treatment of KS.

## Figures and Tables

**Figure 1 cancers-13-05927-f001:**
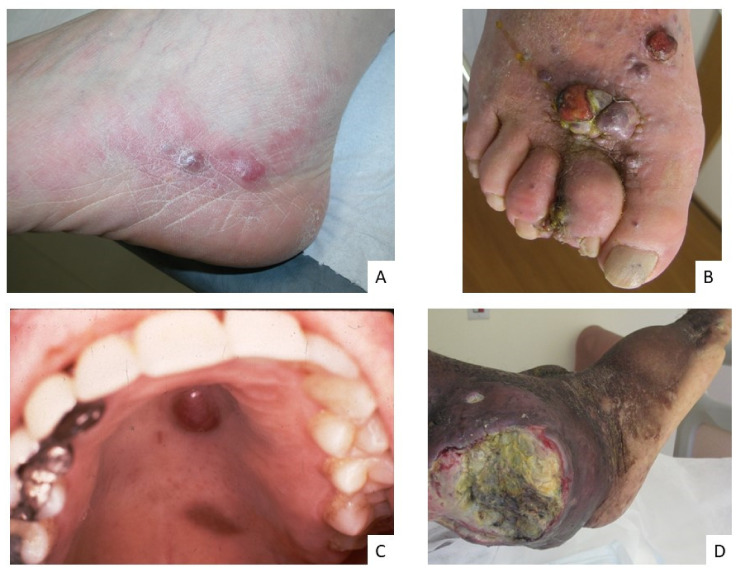
Examples of KS clinical lesions. (**A**) red and pink papules in a patient with classic KS (early stage). (**B**) nodular lesions on the foot in a patient with a more agressive form of classic KS. (**C**) violin macule and nodule of the palate in a patient with AIDS-KS. (**D**) voluminous ulcerative tumor in a patient with an anaplastic form of endemic KS.

**Figure 2 cancers-13-05927-f002:**
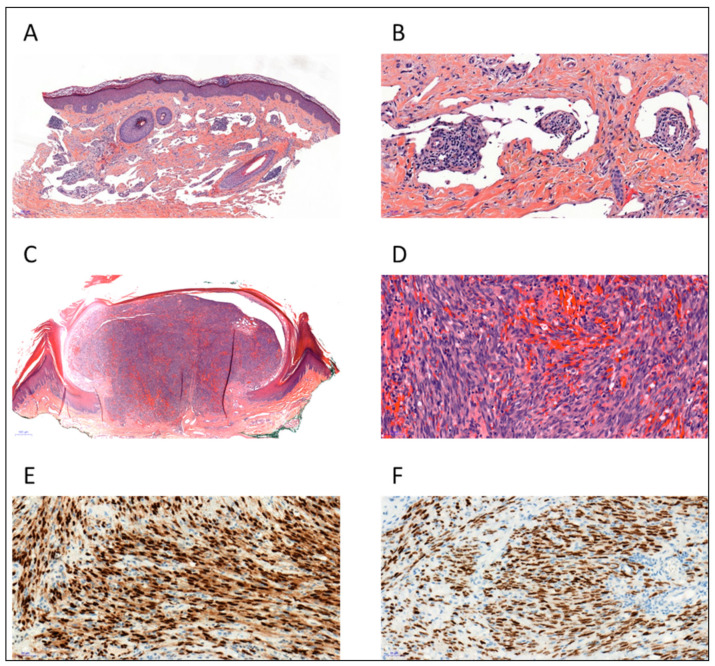
Histopathology of Kaposi sarcoma (KS). (**A**) Patch-stage KS showing dissection of dermal collagen by ectatic vascular spaces (H&E, ×50). (**B**) Small native vessels protrude into abnormal vascular spaces forming the promontory sign, admixed with lymphocytes and plasma cells (H&E, ×200). (**C**,**D**) Nodular-stage KS showing fascicles of spindled cells with slit-like vascular channels containing erythrocytes (H&E ×20 and ×200). (**E**) The spindled tumor cells are positive for ERG (ERG antibody, ×200) (**F**) Tumor cell nuclei demonstrate immunoreactivity for HHV-8 (LNA-1 antibody, ×200).

**Figure 3 cancers-13-05927-f003:**
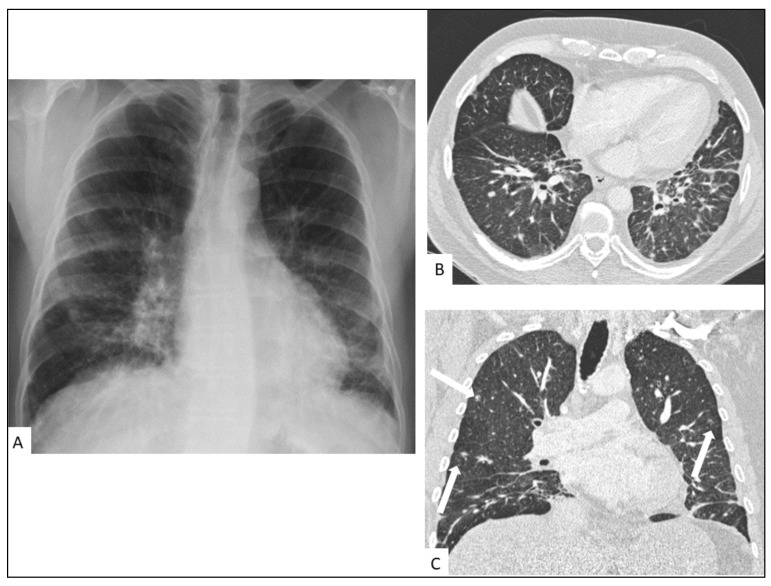
Pulmonary involvement in a 47-year-old male with AIDS-KS. Chest radiography (**A**) shows perihilar infiltrates in the mid and lower lung zones. Chest CT shows bilateral linear and nodular peribronchovascular interstitial thickening with symmetrical distribution in axial (**B**) and coronal planes (**C**) associated with ill-defined nodules (arrows) and interlobular septal thickening.

**Figure 4 cancers-13-05927-f004:**
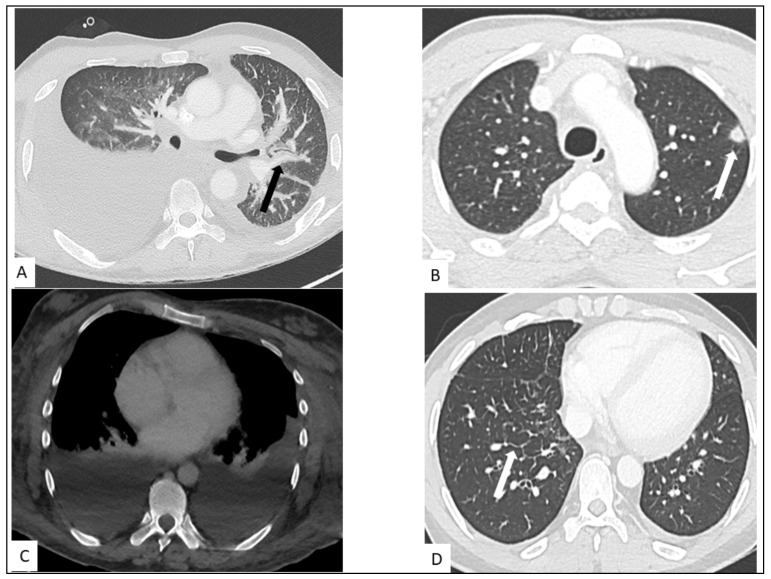
Thoracic involvement in KS. (**A**) Highly abundant right pleural effusion and low abundance left pleural effusion with peribronchovascular thickening (arrow). (**B**) Nodule with halo sign in the left superior lung (arrow). (**C**) Bilateral pleural effusion on contrast-enhanced CT-scan. (**D**) Interlobular septal thickening (arrow) in a secondary pulmonary lobule of middle and lower lungs.

**Figure 5 cancers-13-05927-f005:**
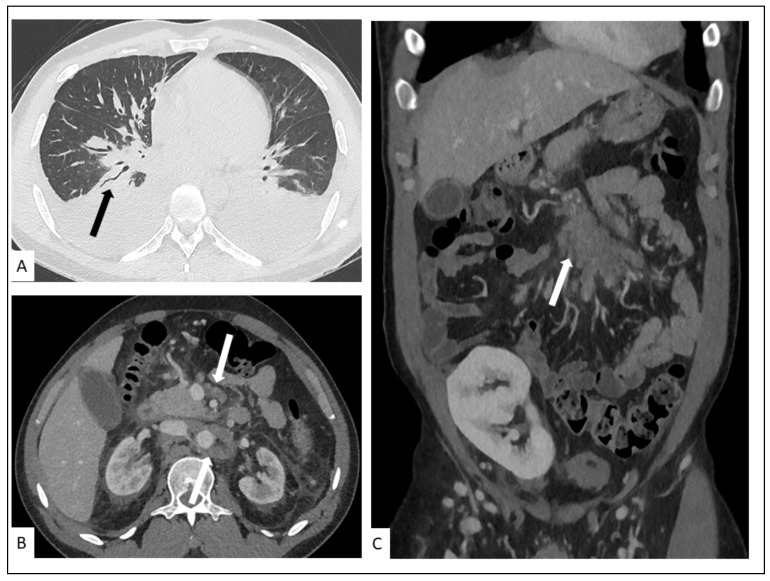
A 27-year-old man, three months after renal transplant. Chest CT-scan showing (**A**) bilateral pleural effusion, peribronchovascular thickening in the lower lungs (arrow) with irregular narrowing of the bronchial lumen. Contrast-enhanced abdominal CT showing in axial (**B**) and coronal (**C**) planes soft-tissue infiltration (arrows) of pancreatic and retroperitoneal lymphadenopathy (arrows) due to pathologically proven KS.

**Figure 6 cancers-13-05927-f006:**
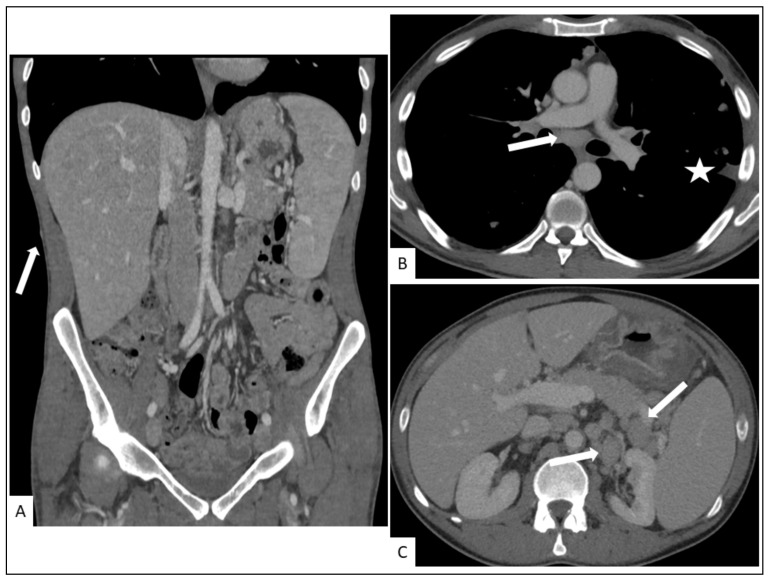
A 29-year-old man with cutaneous, hepatosplenic and lymph nodes involvement in KS. (**A**) Coronal plane contrast-enhanced abdominal CT showing hepato-splenomegaly as well as cutaneous and subcutaneous nodules (arrow). Contrast-enhanced chest CT showing enlarged subcarinal lymph nodes ((**B**), arow), irregularity of the pleural surface ((**B**), asterisk) and abnormally enlarged lymph nodes in the retroperitoneum ((**C**), arrow) due to histologically confirmed KS.

**Figure 7 cancers-13-05927-f007:**
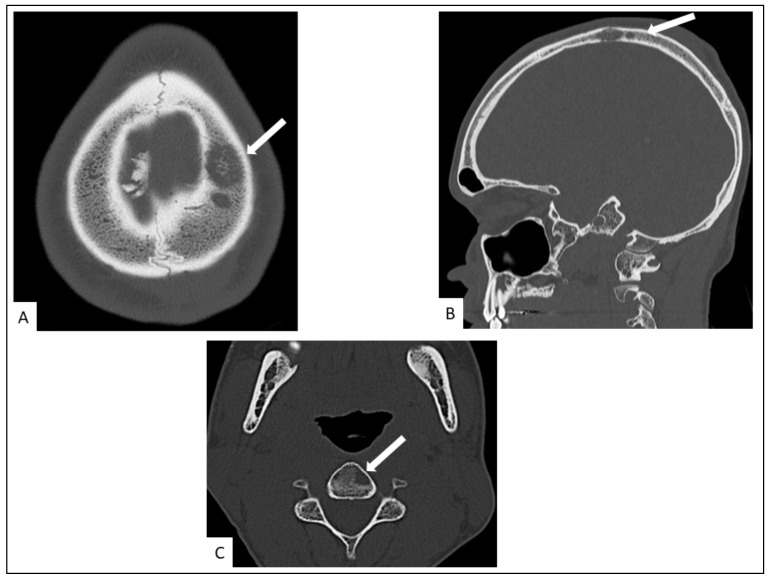
Head CT in a 34-old-year man. It shows lytic left parietal lesion in axial (**A**) and sagittal (**B**) planes and a left vertebral osteolytic cervical lesion (C3) in the axial plane ((**C**), arrow).

**Figure 8 cancers-13-05927-f008:**
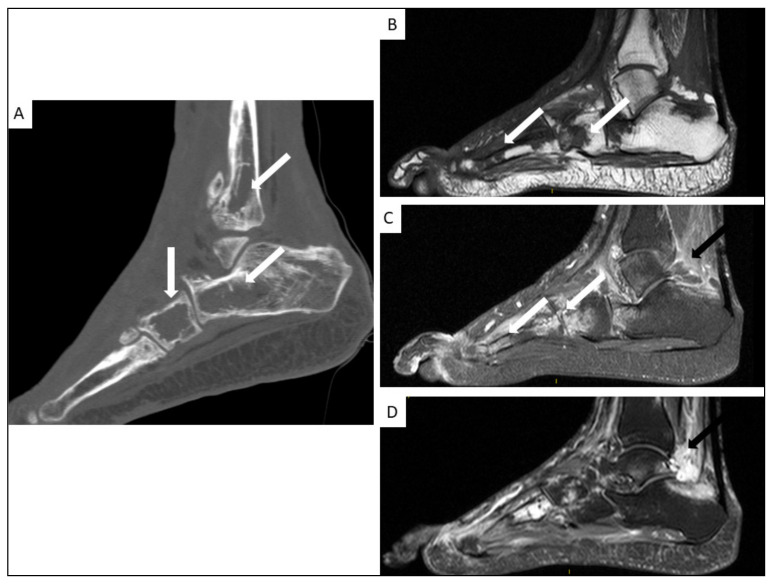
Foot CT-scan (**A**) and MRI (**B**–**D**) of a 38-year-old man with AIDS-KS. Coronal plane CT (**A**) showing multiple osteolytic lesions (arrows), with MRI showing hypo-intensity on T1-weighted images ((**B**), white arrows), hyper-intensity on T2 weighted images ((**C**), white arrows) and important soft tissue signal enhancement after gadolinium injection ((**D**), black arrows) in sagittal slices.

**Figure 9 cancers-13-05927-f009:**
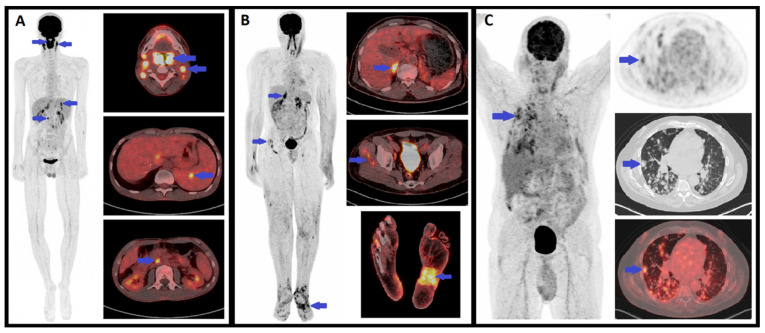
Examples of positive FDG-PET in three patients with KS. (**A**) Maximal intensity projection (MIP) PET and axial plane PET/CT fusion images in a 20-year-old male with iatrogenic immunosuppressive KS (liver transplant) showing involvement of palatine tonsils, cervical and abdominal lymph nodes and the spleen. (**B**) MIP PET and axial plane PET/CT fusion images in a 57-year-old male with endemic KS showing involvement in the right adrenal gland, both feet (skin and subcutaneous involvement) and possibly of the right thigh muscles (not histologically confirmed). (**C**) MIP PET and axial plane PET, CT and PET/CT fusion images in a 58-year-old male with iatrogenic immunosuppressive KS (heart transplant) showing diffuse nodular lung involvement.

**Table 1 cancers-13-05927-t001:** The modified AIDS Clinical Trials Group staging of AIDS-related KS, adapted from Krown, et al. [8].

“TIS” Staging of KS	Good Risk (T0)	Poor Risk (T1)
Tumor	Confined to skin and/or lymph nodes, or minimal oral disease	Tumor-associated oedema or ulceration, extensive oral KS, gastrointestinal KS or KS in other non-nodal viscera
Immune status	CD4 cell count > 150/mm^3^	CD4 cell count < 150/mm^3^
Systemic illness	Karnofsky Performance Status > 70	Karnofsky Performance Status < 70 or other HIV-related illness

**Table 2 cancers-13-05927-t002:** Proposed staging work up, adapted from Lebbe, et al. [11].

Items	Classic/Endemic KS	AIDS-Associated KS	Iatrogenic KS
Clinical examination	++	++	++
HIV serology	++	++	++
Standard blood test	++	++	++
HHV-8 viremia	−	+	+
CD4 count	−	++	+
Histology	++	++	++
Total body CT-scan	+/−	+	+
Bronchoscopy	+/−	+/−	+/−
Esophago-gastro-duodenoscopy	+/−	+/−	+/−

NOTES. − Usually not; + usually yes; ++ mandatory; +/− according to symptoms.

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
