# Peer review of "Current and Future Tools for Diagnosis of Kaposi’s Sarcoma"

_cancers, 2021, doi:10.3390/cancers13235927_

Round 1

Reviewer 1 Report

The authors review a topic of exceptional importance and that presents a unique challenge in the practice of oncology, i.e., reliably and precisely staging and monitoring KS in clinical care. They also discuss highly innovative technologies that may solve the KS challenges, and in doing so, may open new horizons in detection and measurement of malignant lesions in other cancers. The value of this review cannot be overstated!

However, there are several aspects of the review that need editing:

  • The authors seem to interchange the term “epidemiological” with “clinical” forms of KS. For clarity, it is important that these two are not interchanged. Epidemiological forms refer to etiology/geographical patterns (i.e., endemic, classic, iatrogenic, epidemic) while clinical phenotypes refer to anatomical patterns (i.e., mucocutaneous, lymphadenopathy, visceral, or lymphedema).
  • The authors refer to skin as the most common type of KS and pathognomonic. This is a rather broad and misleading statement because, (A) In some populations such as children with HIV infection, lymphadenopathy KS is more common that skin lesions; (B) although a diagnosis of KS can be made clinically based on the appearance of skin lesions, this is a little misleading in many cases since there are numerous differentials of such skin lesions as demonstrated by Amerson E et al. (Accuracy of Clinical Suspicion and Pathologic Diagnosis of Kaposi Sarcoma in East Africa. J Acquir Immune Defic Syndr 2016; 71(3): 295-301.) Therefore, it is rather hyperbole to consider any skin lesion pathognomonic of KS.
  • In oncology, staging (i.e., the anatomical extent of spread of cancer) and risk stratification (i.e., the composite of clinical and pathological factors that predict prognosis with a specified therapy) are related but distinctive concepts. In their review, the authors seem to conflate the two concepts. It will improve the strength and clarity of the review if the authors distinguish staging from risk stratification wherever relevant the manuscript.
  • Although the authors have done their best to present the manuscript in clear English, there are still several instances where less appropriate terms are used, thus altering the meaning. The manuscript may benefit from minor English language editing.
  • There two different fonts used in parts of the abstract.

Reviewer 2 Report

The manuscript reviews current and future tools for diagnosis of Kaposi’s sarcoma (KS).

Major comments

#1:

I don’t feel the importance to review current and future tools for diagnosis of KS, because the authors don’t discuss the importance in the Introduction section.

On the other hand, the authors mention in Conclusion section (lines 430-433) that “KS is an opportunistic tumoral process which in most cases is an indolent disease and does not need specific treatment. However, in some instances, KS may be locally aggressive or disseminated, requiring a reliable assessment of the disease stage before choosing the appropriate therapeutic option.”

But, it is not enough. At first, the authors must discuss the importance in the Introduction section. Otherwise, future readers may not be interested in the review.

#2:

To detect the mucosal (visceral) KS, esophagogastroduodenoscopy and/or colonoscopy seems to be useful. Therefore, please discuss about the tools.

Minor comments

#3:

Abstract section: please replace FDG-PET and LDI with “[18F]-fluoro-deoxy-glucose positron emission tomography” and “laser Doppler imaging”, respectively.

#4:

Figure 1: If possible, please also indicate clinical lesion of mucosal (visceral) KS.

#5:

Line 410: please replace “laser Doppler imaging” with “LDI”.

#6:

Line 417: please delete “(PCA)”.

#7:

Line 436: please correct “PDL-1” to “PD-L1”.

Round 2

Reviewer 2 Report

Thanks for the revisions. In the cover letter of the authors, wrong line numbers are indicated. Not "L100-110" and "L166-168", but "L90-99" and "L151-153", respectively. But, the revisions are satisfactory.